



# A case study of the day-to-day occurrence of plasma irregularities in low-latitude ionosphere from multi-satellite observations

**Weihua Luo[1], Chao Xiong[2], Zhengping Zhu[1], Shanshan Chang[1], and Xiao Yu[3]**

[1] College of Electronic and Information Engineer, South-Central University for Nationalities, 430074 Wuhan, China.

[2] Helmholtz Centre Potsdam, GFZ German Research Centre for Geosciences, Telegrafenberg, 14473 Potsdam, Germany.

[3] China Research Institute of Radiowave Propagation, Qingdao, China

Correspondence: Weihua Luo (whlu@whu.edu.cn)

## Abstract

Day-to-day variability of the occurrence of plasma irregularities in low-latitude ionosphere is still an open issue. In this study, we report the occurrence of post-sunset plasma bubbles and blobs detected by the First satellite of the Republic of China (ROCSAT-1) in the same longitude sector (170°E) on two successive days, under geomagnetically quiet and disturbed conditions, respectively. Multi-Low Earth orbit (LEO) missions, like the Defense Meteorological Satellite Program (DMSP) F13 and F15, the Gravity Recovery and Climate Experiment (GRACE) and the Challenging Mini-satellite Payload (CHAMP) satellites are used to study the preferable conditions for the occurrence of plasma bubbles and blobs. The observations from the CHAMP and GRACE show that the Equatorial Ionization Anomaly (EIA) was enhanced significantly before the occurrence of plasma irregularities on both two successive days. We suggest that the enhancement of post-sunset eastward electric field is the most important factor for the day-to-day development of the plasma irregularity in equatorial and low-latitude ionosphere. In addition, the meridional neutral wind plays an important role in the occurrence of low-latitude plasma blobs.

Keywords: low-latitude ionosphere; day-to-day variability; plasma bubble; plasma blob;

Key points:



➢ Low-latitude plasma bubbles and plasma blobs were detected by ROCSAT-1 on
two successive days in 170°E region under quiet and disturbed conditions,
respectively
➢ Equatorial Ionization Anomalies were enhanced significantly before the
occurrence of the plasma irregularities on the two successive days
➢ Enhancement of the post-sunset eastward electric field plays an important role in
the day-to-day occurrence of low-latitude plasma bubbles and plasma blobs

## 1. Introduction


The occurrence of plasma irregularities, including plasma density depletion
(plasma bubble)/equatorial spread F (ESF) and plasma density enhancement (plasma
blob), is one of particular phenomena after sunset in equatorial and low-latitude
ionosphere, which display remarkable variations with local time, day-to-day, season,
longitude, solar cycle, geomagnetic conditions and so on (Kelley, 2009). In general,
the development of plasma bubble is thought to be initiated by the Rayleigh-Taylor
(R-T) instability at the bottomside of F-region in equatorial ionosphere, the bubbles
move upward and penetrates into topside ionosphere under the effect of $\mathbf{E} \times \mathbf{B}$. Based
on multiple observations and numerical simulations, many reports have denoted that
the pre-reversal enhancement (PRE) of the zonal electric field or vertical drift, neutral
wind, solar flux, gravity wave (GW), Large Scale Wave Structure (LSWS) may affect
the initiation of the plasma bubbles (e.g. Kelley, 2009; Tsunoda, 2010; Tsunoda et al.,
2018). Up to date, the factors leading to the day-to-day occurrence of plasma bubbles
are still an enigma.
The PRE vertical drift/zonal electric field is thought as a basic requirement for
the development of post-sunset plasma bubble. Many studies have demonstrated that
there is a "threshold" of vertical drift for the initiation of R-T instability (Huang, 2018,
and references therein). However, the bubble may occur on the days when the drifts
are lower than the "threshold", or be absent on the days when the drifts beyond the
"threshold" (e.g. Fejer et al., 1999; Lee et al., 2005).
Some studies also indicated that gravity waves, seeding R-T instability, may
affect the day-to-day variability of plasma irregularity (e.g. Tsunoda, 2005; Aswathy
and Manju, 2017; Tsunoda et al., 2018). Tsunoda et al. (2018) suggested that the
controlling driver for bubble development may be related with the coupling between



the lower atmosphere and ionosphere, not vertical drift, such as  the gravity waves
(GW) or LSWS, planetary waves (PW), which are considered to last for several hours,
from late afternoon to dusk and not easy to detect using ground-based observations
(Tsunoda, 2005). The numerical simulations have shown that the forcing from lower
atmosphere (e.g. GW) can initiate the R-T instability as a seed (Huang and Kelley,
1996; Krall et al., 2013), whether the growth of R-T instability can be amplified at
later local time is not clear.
Thermospheric neutral wind is another candidate for the day-to-day variability of
the plasma bubbles. Many experiments and numerical simulations have confirmed
that the meridional and vertical neutral wind play an important role in the evolution of
R-T instability and bubbles, depending on the directions of wind (Maruyama, 1988;
Mendillo et al., 2001; Krall et al., 2009b; Sekar et al., 1994).There is no convincing
evidences that meridional winds at dusk exert a strong influence on day-to-day post-
sunset ESF onset (Mendillo et al., 2001).
Moreover, since the plasma blob was firstly reported by Watanabe and Oya
(1986), the mechanism for the generation of plasma blob has not been well
understood until now. Some studies indicated that the occurrence of plasma blob may
be related with the plasma bubbles, occurred earlier at lower latitudes, due to the
polarization electric field (Le et al., 2003; Huang et al., 2014; Yokoyama et al., 2007)
and/or neutral wind (Krall et al., 2009a; Martinis et al., 2009; Park et al., 2003; Luo et
al., 2018). Some studies showed that the plasma blob may generate independent on
the occurrence of plasma bubble (Choi et al., 2012; Kil et al., 2011), and they
contribute the formation of plasma blob to the traveling ionospheric disturbances (Kil
et al., 2015; Martinis et al., 2009). Other studies proposed that the development of
plasma blob may result from multi-mechanisms, because the blob can occur under the
presence or the absence of the plasma bubble (Klenzing et al., 2011; Haaser et al.,
92  2012).

However, previous studies on the possible factors for the day-to-day variability
of plasma irregularities were statistical or from observations on a day or separate days.
Thus, on the basis of multiple observations, more case studies in some successive
days under similar and/or different geophysical conditions should be carried out to
understand the day-to-day characteristics of plasma bubbles and blobs, and to
investigate the possible factors or mechanisms for the day-to-day occurrence of





99 plasma bubbles and blobs. In this paper, we report the occurrences of plasma bubbles

100 and plasma blobs detected by the First satellite of the Republic of China (ROCSAT-1)

101 in 170°E longitude sector on two successive days, under quiet and disturbed

102 conditions, respectively. Observations from Defense Meteorological Satellite Program

103 (DMSP) F13 and F15 satellites, Challenging Mini-satellite Payload (CHAMP) and

104 Gravity Recovery and Climate Experiment (GRACE) are also presented to discuss the

105 possible factors on the day-to-day occurrence of plasma bubbles and plasma blobs.

106 **2. Data descriptions**

107   The DMSP is a series of satellites that fly in near-circular orbits (inclination:

108 98.7°) at about 840 km. The Special Sensor-Ions, Electrons, and Scintillation (SSIES)

109 on board the satellites measures the ion and electron densities, temperatures and drifts.

110 All DMSP satellites fly in Sun-synchronous orbits near either the 0600-1800 local

111 time (LT) or the 0930-2130 LT meridians (Rich and Hairston, 1994). In this study, we

112 use the data from the DMSP F13 and F15 satellites, whose orbit is on the 0600 LT

113 and 0900 LT meridian, respectively.

114   The Ionospheric and Plasma Electrodynamics Instrument (IPEI) on board the

115 ROCSAT-1, which was launched to a circular orbit on March 1999 with a 35°

116 inclination orbital plane around 600 km, measures in-situ ion density, temperature and

117 ion composition in the low-to-middle-latitude ionosphere (Yeh et al., 1999).

118   The CHAMP satellite was launched into an almost circular, near-polar orbit

119 (inclination:87.3°) on 15 July, 2000, with an initial altitude of about 454 km. In 2003,

120 the orbit decayed to about 400 km (Reigber et al., 2002). The Planar Langmuir Probe

121 (PLP) on board the satellite measures the in-situ electron density and temperature per

122 15 seconds. The GRACE mission, including two spacecraft GRACE-A and GRACE-

123 B, was launched into a polar-orbit (inclination:89°) on 17 March 2002, at an initial

124 altitude of about 490 km (Tapley et al., 2004).

125   The electron density from PLP and the electron density derived from the K-band

126 ranging (KBR) system between the GRACE two spacecraft (Xiong et al., 2010) have

127 been validated by comparison with ground-based measurements at Jicamarca

128 (McNamara et al., 2007) and European Incoherent Scatter radar (EISCAT), Millstone

129 hill and Arecibo radars (Xiong et al., 2015), respectively. The electron densities from

130 CHAMP and GRACE provide good opportunities to investigate the latitudinal





characteristics and variations of ionosphere at low- and middle latitudes, e.g., the
Equatorial Ionization Anomaly (EIA) (e.g. Xiong et al., 2013).

### 3. Results

#### 3.1 Observations from multi-satellites/ROCSAT-1

#### 3.1.1 Plasma bubbles and plasma blobs detected by ROCSAT-1

Figure 1  presents the plasma bubbles in 170°E  region recorded by ROCSAT-1
during 0940-0950 Universal Time (UT) on 17 August (left panel, I) and during 0949-
0952 UT 18 August (right panel, II) 2003, respectively. Variations of three
components of ion velocity, ion compostion, and ion temperature along the satellite
trajectories are also displayed in the figure, and magnetic latitude, geographic
longitude, UT, LT are noted at the bottom of figure. $V_{meridional}$ and $V_{zonal}$ are the two
components perpendicular to the field line in the meridional (upward) and zonal
(eastward) direction, respectively. $V_{parallel}$ is field-aligned velocity.
As shown in Figure 1, in the same longitudinal  region, plasma bubbles were
recorded by ROCSAT-1 in two successive days, 17 and 18 August 2003.
On 17 August, in 180°E, the background density was about  $2.75 \times 10^5$ cm$^{-3}$. At
7.46°N, the density decreased to about $9.4 \times 10^4$ cm$^{-3}$, and reached a minimum at 6.9°N
(181°E), with the magnitude of $3.64 \times 10^4$ cm$^{-3}$. The density depletions were about
65.8% and 86.8%, respectively.
On 18 August, the density was about $1.71 \times 10^5$ cm$^{-3}$ at 4.59°N, and $1.58 \times 10^5$ cm$^{-3}$
at 4.14°N. The background density was about $6.12 \times 10^5$ cm$^{-3}$, which means the
density depletions reached about 72.1%  and 74.2%, respectively. At 1.69°N
(180.08°E), the density reached a minimum of about $6.71 \times 10^2$ cm$^{-3}$, the background
density was about $3.5 \times 10^5$ cm$^{-3}$. The magnitude of density depletion exceeded 90%.
At 1.52°N (180.37°E), the density was about $1.35 \times 10^4$ cm$^{-3}$, which means the
depletion was about 96.1%.
On these two days, the bubbles moved poleward, westward and upward.
However, there were differences between the bubbles on the two successive days. On
17 August, the $O^+$ composition inside the bubble showed increase or decrease, while
$O^+$ composition inside the bubble on 18 August decreased.





Similar with Figure 1, Figure 2 presents the plasma blobs in 170°E region
recorded by ROCSAT-1 during 1118-1129 UT on 17 August (left panel, I) and during
1128-1135 UT on 18 August (right panel, II) 2003, respectively.
As shown in Figure 2, in 170°E region, plasma blobs were recorded by
ROCSAT-1 on two successive days, 17 and 18 August 2003, occurring after the
earlier occurrences of the plasma bubbles.
On 17 August, in Region 1, at -6.58°N, the ion density increased to $2.35 \times 10^5$ cm$^-$
$^3$, the background density was about $1.0 \times 10^5$ cm$^{-3}$, which means the enhancement
reached about 135%. In Region 2, the ion density was about $2.22 \times 10^5$ cm$^{-3}$ at -3.81°N
and $2.48 \times 10^5$ cm$^{-3}$ at -3.5°N, which enhanced about 122% and 148% with respect to
the background, respectively. In Region 3, the density was about $1.94 \times 10^5$ cm$^{-3}$ at
0.86°N and $1.96 \times 10^5$ cm$^{-3}$ at 0.98°N, respectively. The background density was about
$1.38 \times 10^5$ cm$^{-3}$, the enhancements were about 40.6% and 42%, respectively.
In Region 4, the background density was about $1.86 \times 10^5$ cm$^{-3}$ , the density
decreased to about $1.41 \times 10^5$ cm$^{-3}$ at 4.31°N, which means the depletion was 24.7%.
On 18 August, in Region 1, the ion density increased to $5.48 \times 10^5$ cm$^{-3}$ at -
16.27°N and $5.09 \times 10^5$ cm$^{-3}$ at -15.96°N, the background density was about $4.38 \times 10^5$
cm$^{-3}$, which means the enhancement reached about 25.1% and 16.2%, respectively. In
Region 2, the ion density was about $6.97 \times 10^5$ cm$^{-3}$ at -14.15°N and $6.08 \times 10^5$ cm$^{-3}$ at -
14.62°N, which enhanced about 51.2% and 24.2% with respect to the background,
with the magnitude of $4.61 \times 10^5$ cm$^{-3}$, respectively. In Region 3, the density was about
$7.16 \times 10^5$ cm$^{-3}$ at -12.1°N and $7.01 \times 10^5$ cm$^{-3}$ at -12.02°N, respectively. The
background density was about $4.68 \times 10^5$ cm$^{-3}$, the enhancements were about 53% and
49.8%, respectively.
In Region 4, there was small density disturbance, with the magnitude of about
10.2%. The drifts, composition and ion temperature show obvious fluctuations.
On 17 August, the O$^+$ composition inside the blob showed increase, while O$^+$
composition inside the bubble on 18 August decreased.
**3.1.2 Tracks and observations of multi-satellites**
To investigate the evolution of the plasma bubbles and blobs detected by
ROCSAT-1, we also study the observations from other satellites in the same





longitudinal region. In Figure 3, we display the observations from ROCSAT-1,
DMSP, GRACE and CHAMP satellites in 130-190°E region, and also the tracks of
the satellites during 17-18 August 2003, at four different UT periods. The plasma
bubbles and blobs recorded by satellites are also given in the figure, the red and green
short lines represent the density enhancements (plasma blobs) and density depletions
(plasma bubbles) encountered by the satellites, respectively.
From the Figure 3, we notice that the plasma bubbles were also detected in same
region on 17 August by GRACE and DMSP at different altitudes, besides the
ROCSAT-1, respectively. As displayed in Figure 3, the plasma bubbles were detected
in 180°E regions on two successive days. On these two days, the bubbles moved
westward, and after about 100 minutes, the plasma blobs were recorded in 170°E
regions.

**3.2 Observations from DMSP**

Figure 4 shows the variations of ion density from DMSP F13 and F15 satellites
along the satellite trajectories during 15-19 August 2003. The variations of longitude
are also displayed in dashed lines.
It can be seen that the density around 0616 UT and 1026 UT at DMSP altitude
increased significantly on 18 August 2003. On 17 August, it is interesting that the
density showed obvious variations during different periods. The density around 0631
UT observed by DMSP F13 satellite was close to that around 0645 UT of 16 August.
After more than 2 hours, around 0858 UT, the density in low-latitude region was
larger obviously than that on other days.

**3.3 EIA variations from CHAMP and GRACE**

Figure 5 displays the variations of electron density along the satellite trajectories
in low-latitude ionosphere from CHAMP and GRACE observations in 170°E sector
during 1-22 August 2003, respectively. The satellite tracks are also shown in the
figure. The local times measuring from CHAMP and GRACE were around 1900-2000
LT (after local sunset) and 2100-2130 LT, respectively.
In is seen that the densities on 17 and 18 August were larger above the crests and
smaller at the trough with respect to that on other days of August 2003. On 17 and 18
August, EIA crests moved poleward, both at CHAMP altitude and GRACE altitude.
Furthermore, similar with Luo et al. (2017), we calculate the EIA strength (Crest-to-



Trough Ratio, CTR) and asymmetry (ASY), which are calculated as *CTR=(N+S/2T)*,
*ASY=(N-S)/((N+S)/2)*, respectively, shown in Figure 6. *N(S)* represents the density
above the northern (southern) crest, *T* represents the density at the trough. When the
EIA was not well developed, CTR is set as 1 and ASY is taken as 0, respectively.

The EIA stengths were enhanced  significantly on 17 and 18 August, manifesting

as the remarkable decrease of density at the trough and the increase of densities above
the crests. CTR reached 50 on 17 August and 710 on 18 August, respectively. During
the two days, especially on 18 August, EIA asymmetries at CHAMP altitude and
GRACE altitude became more remarkable than that on 16 August, and 19 August.
**3.4 Variations of [O/N2] ratio**

Figure 7 displays the variations of Global Ultraviolet Imager (GUVI) [O/$N_2$]

ratio during 16-19 August 2003.

Around the magnetic equator, the [O/$N_2$] ratio increased on 17 and 18 August,

with respect to that on 16 August and 19 August.
**4 Discussion**

The occurrence of post-sunset plasma irregularities in equatorial and low-latitude

ionosphere displays remarkable characteristics varying with day-to-day, season, solar
cycle and so on. Generally, the plasma irregularity is preferable to occur during spring
and fall equinox, under high solar flux (Aarons, 1993). Geomagnetic disturbance may
promote or inhibit the occurrence of plasma irregularity (Martinis, 2005).

In linear theory of the development of R-T instability and plasma irregularity,

background electric field and neutral wind are crucial factors for the initiation of R-T
instability. In dipole coordinate system (*q,s,l*), the linear growth rate (*γ*) of R-T
instability can be described as (e.g. Basu, 2002; Luo et al., 2013)
$$\gamma = \frac{1}{\eta_0 L_{nq}}(\frac{g}{\nu_{in}} + \frac{E_s}{B} - U_q + \frac{\nu_{in}}{\Omega_i}\frac{E_q}{B} + \frac{\nu_{in}}{\Omega_i}U_s) - \beta ,$$    (1)

Where *Lnq* represents the scale length of vertical density gradient, $\nu_{in}$ is the ion-

neutral collision frequency, $\Omega_i$ is the gyrofrequency of ions, $E_s$ and $E_q$ represent the
background zonal and vertical electric field, *Uq* and *Us* represent the vertical and
zonal wind, respectively.



As shown in Equation (1), the zonal electric field can strengthen or decrease the
growth rate depending on the direction, which is thought as a basic requirement for
the development of plasma irregularity after sunset. The growth rate can be affected
directly by the vertical wind, which has been discussed by some studies (e.g. Sekar
and Raghavarao, 1987; Raghavarao, 1999). Numerical simulations also have indicated
that the nonlinear evolution of plasma bubbles would be amplified by vertical wind
(Sekar et al., 1994; Krall et al., 2013; Yokoyama et al., 2019). It is well known that
eastward electric field and downward wind would favor the development of R-T
instability, which may play an important role in the day-to-day variability of the
plasma irregularity.
In addition, the zonal electric field and meridional/vertical wind can affect the
electrodynamics in equatorial and low-latitude ionosphere, such as the formation of
EIA. Many studies have demonstrated that the variations of EIA, including EIA
strength and asymmetry, are generally related with zonal electric field and meridional
wind (Mendillo et al., 2001;Lin et al., 2005; Balan et al., 2018; Khadka et al., 2018).
Thus, we can conclude the characteristics of zonal electric field and meridional wind
in F-region from the EIA variations.
In this study, one interesting observation is that quite prominent plasma
irregularities were detected during 17-18 August 2003, the relative low season of the
occurrence of plasma bubbles, the density depletion or enhancement were not
detected in same longitudinal sector on other days. Figure 8 presents an example for
the variations of satellite tracks (top panel) and ion density (bottom panel) along the
close ROCSAT-1 satellite trajectories on successive days.
The plasma irregularities were recorded on 17 and 18 August, and were absent
on other days. The detailed density variations in 170°E region from ROCSAT-1 on 16
August and 19 August can be find in "Supplement".
The other interesting result is that the plasma bubbles and blobs were detected on
two successive days under different geophysical conditions, respectively. The
geophysical conditions during the occurrences of the plasma irregularities are given in
Figure 9. During 16-20 August, 2003, the variations of $Dst$ index (a), $K_p$ index (b),
solar wind speed $Vsw$ (c), the south-northern component of Interplanetary Magnetic
Field (IMF) $B_Z$ (d), and the Interplanetary Electric Field (IEF) $E_y$ ( $E_y = V_{sw} \times B$ )(e),
and also the variations of Prompt Penetration Electric Field (PPEF) and quiet plus





penetration electric field derived from the real-time model of the ionospheric electric
fields (Manoji and Maus, 2012) at 170°E sector (f) are displayed, respectively. The
red line represents the penetration electric field calculated from the IEF $E_y$ and
transfer function (Manoji and Maus, 2012), the green line represents the quiet
background electric field plus the penetration electric field.

The storm sudden commencement (SSC) of the 17-20 August 2003 storm was at

1421 UT on 17 August. On 17 August, before the SSC, the $Dst$ index did not show
sudden variations and  $Kp$ indices  were not more than 1. On 18 August, the $Dst$ index
dropped to a minimum (-148 nT) around 1600 UT, and $K_p$ indices were no less than 6.
After the SSC, $E_y$ was westward and turned to east at about 1800 UT on 17 August.
From Figure 8, it can be noted that the plasma irregularities detected by ROCSAT-1
occurred on a geomagnetically quiet day (17 August) and a disturbed day (18 August),
during main phase of the storm, respectively.

On 17 August, as shown in Figure 8, before the SSC, the $Ey$ was westward and

the PPEFs were very small, which means that the background electric field may not
be affected by the factors from the upper, such as PPEF.

On 18 August, the IEF $E_y$ was always eastward and reached a maximum at 0800

UT, with a value of about 7.8 mV/m. The IEF would partially penetrate into the
ionosphere as prompt penetration electric field (PPEF) with an efficiency of ~5 to 10%
(Verkhoglyadova et al., 2008), with sudden variations of $Bz$. The IMF $Bz$ was
southward with sudden variations during 0000-0900 UT. It can be seen from Figure
8(f) that the penetration electric field was eastward during 0300-0800 UT, with the
maximum of about 0.18 mV/m at 0530 UT. It means that the PPEF would affect the
background electric field during 0300-0800 UT (local daytime), and the EIA may be
enhanced due to PPEF.

As shown in Figures 5 and 6,  on 17 and 18 August, the EIA derived from the

CHAMP and GRACE observations showed remarkable enhancement. EIA strength is
directly related with the zonal electric field. The enhanced eastward electric field,
leading to a "super fountain" effect, would drive the EIA crests move toward the
poles to higher latitudes, the density increase greatly above the crests while decreases
near the magnetic equator (Lin et al., 2005; Lu et al., 2013; Balan et al., 2018). On the
other hand, equatorward neutral wind may also strengthen the EIA (Lin et al., 2005;
Balan et al., 2018), driving the crests move to the equator.



From Figure 5, we can notice that EIA crests moved toward the poles on 17 and
18 August, with the remarkable increases of density above the crests and decrease
near the magnetic equator. In addition, from Figure 4, the observations from DMSP
F13 and F15 showed that the density on 17 August at DMSP altitude (840 km) was
close to that on 16 August, and increased around 0858 UT with respect to that on 16
August, it means that the eastward electric field was strengthened during that period,
which driven the density from the lower ionosphere to the higher altitude. Thus, it can
be concluded that the density variations in EIA regions and motions of the crests
indicated that the eastward electric field (PRE vertical drift) was enhanced after 0630
UT (around local sunset, before the occurrence of plasma bubbles) on 17 August in
170°E sector, leading to the EIA enhancement. As discussed before, the factor
causing the increase of eastward electric field was not from the upper, which means
that the EIA enhancement may be not associated with the storm.
Similar with the situation on 17 August, on 18 August, it can be concluded that
the eastward electric field around local sunset was amplified before the occurrence of
the plasma bubbles, which may be partially related with the PPEF.
Moreover, the EIA asymmetry on 17 and 18 August also showed obvious
variations with respect to that on 16 August. On 17 August, EIA became more
asymmetric with respect to that on 16 August, and the asymmetry on 18 August
became more remarkable. The EIA asymmetry is generally related with the
meridional neutral wind(Mendillo et al., 2001; Lin et al., 2005; Balan et al., 2018;
Khadka et al., 2018 ), which modify the distribution of ionization with respect to the
magnetic equator. Furthermore, it can be noted from Figure 5 that the characteristics
of EIA asymmetry were different at different altitude. At CHAMP altitude, the ASY
was positive, which means that the density above northern crest was larger that above
the southern crest, and at GRACE altitude, the density above the nothern crest became
smaller than that above the souther crest. The variations of the density above the
crests at the different altitude means that the presence of the meridional wind on 17
and 18 August, with the altitudinal and/or horizontal gradient (Huba and Krall, 2013).
Some reports have indicated that the trans-equatorial meridional wind may suppress
the development of ESF/plasma bubbles (Maruyama, 1988; Mendillo et al., 2001;
Krall et al., 2009b), and the plasma bubble is preferable to occur when the EIA is
symmetric (Lee et al., 2005; Thampi et al., 2008). On 17 and 18 August, though the
presence of the meridional wind, the plasma bubbles were recorded, which means that





the enhancement of eastward electric field is dominant for the occurrence of the
plasma bubbles, whilethe meridional wind is not the dominant factor.
In addition, we notice that the plasma blobs were recorded on these two
suceesive days. On these two days, as shown in Figure 1 and Figure 2, the plasma
blobs were detected about 100 minutes after the occurrences of plasma bubble, and
the variations of ion composition inside the blobs were similar with that inside the
bubbles, which means that the blobs may be associated with the bubbles. Considering
the relationship between the plasma bubble and blob, Huang et al. (2014) proposed
that the polarization electric field due to the occurrence of plasma bubble would lead
to the formation of plasma blob. Observational and numerical results also indicated
that the meridional wind may cause the formation of plasma blob (Krall et al., 2009,
2010; Luo et al., 2018). In this study, it can be speculated that the presence of
meridional wind on 17 and 18 August, from the variations of EIA shown in Figure 5.
As Luo et al. (2018) proposed, under the effects of meridional wind, flowing from
summer to winter hemisphere, in addition to the polarization electric field after the
occurrence of plasma bubbles (Huang et al., 2014), the plasma blob could occur in
winter hemisphere due to the accumulation of plasma in the low-latitude region.
There is another interesting result should be noted, as shown in Figure 7, the
GUVI [$O/N_2$] ratio displayed the increase during 17-18 August in 150-180°E region.
The compositional variability is mainly driven by both vertical and horizontal winds.
The increase of [$O/N_2$] may be related with the downward wind (Rishbeth, 1998; Lin
et al., 2005).  Though the source of vertical wind in equatorial region is still not well
understood (Larsen and Meriwether, 2012), the existences of vertical wind have been
reported at different longitude sector (Biondi and Sipler, 1985; Raghavarao et al.,
1993; Herrero and Meriwether, 1994). Raghavarao et al. (1993) proposed a possible
source of vertical wind in equatorial region. When EIA enhanced, the pressure in EIA
regions would be strengthened, and the enhanced pressure ridges would give rise to a
downward wind in equatorial region and upward winds in crest regions (Raghavarao
et al., 1993).
From Figures 4 and 5, we notice that EIAs were strengthened significantly with
respect to other days in August. The vertical wind may be produced by the enhanced
pressure in equatorial region, and the downward vertical wind would amplify the
development of the R-T instability, and cause the increase of [$O/N_2$] ratio.





Thus, it can be concluded that the factors leading to the occurrence of plasma
bubbles and blobs on 17 and 18 August are attributed to the enhanced eastward
electric field after sunset, the vertical neutral wind due to the strengthen of EIA and
the meridional neutral wind.
As mentioned before, the enhancement of eastward electric field on 17 August
may be not from the upper, the enhancement of eastward electric field on 18 August
may be partially from the PPEF. Other sources for the enhancement are from the
below, such as the LSWS, PW and GW. The wave structure, manifesting itself as the
height oscillations of the bottomside F layer at daytime, becomes amplified towards
post-sunset hours. The vertical and zonal winds, associated with gravity wave
propagating zonally and slant upward, would generate a polarization electric field. A
vertical perturbation wind ( $\Delta U_Z$ ) produces the zonal polarization electric field ( $\Delta Ey$ )
as $\Delta Ey = -\Delta U_Z \times B_0$. Some observations have demonstrated that the propagating LSWS
may generate polarization electric fields and enhance the PRE vertical drift (Fagundes
et al., 1999; Abdu et al., 2015; Ajith et al., 2018; Abdu, 2019), and many studies have
showed the occurrence of plasma irregularities may be related with the LSWS (e.g.
Thampi et al., 2009; Tsunoda et al., 2018).
It can be speculated that the enhancement of eastward electric field on 17 August
may be associated the presence of propagating LSWS or GW. Unfortunately, we
cannot find any evidences to study the presence of LSWS in the ionosphere. The
relationship between the LSWS and the enhancement post-sunset electric
field/vertical drift, and also the occurrence of plasma bubbles need to be further
studied from multiple observations.
In conclusion, the occurrences of the plasma bubbles and blobs in this case on
both two successive days can be concluded as: 1) at first, the PRE vertical drift
(eastward electric field) was enhanced due to the factors from below on 17 August,
e.g. gravity waves, and the PPEF on 18 August, respectively; 2) EIA became stronger
and R-T instability was initiated under the effect of the enhanced post-sunset eastward
electric field, in addition to the downward neutral wind resulting from the
strengthened EIA, leading to the occurrence of the  plasma bubbles; 3) under the
effect of the meridional neutral wind, EIA became more asymmetric, and the plasma
blobs occurred on 17 and 18 August. In a word, the enhancement of post-sunset
eastward electric field is the most important for the occurrence of plasma bubble





under quiet conditions, giving rise to the vertical wind, whatever resulting from the
below or above, such as LSWS or PPEF, even in the low season of the occurrence of
plasma irregularity. The meridional wind may be not the dominant factor for the
occurrence of the plasma bubble, but for the plasma blob.
**5 Conclusions**
In this paper, we present the occurrence of plasma bubbles and blobs in the same
longitude sector (170°-180°E) on the two successive days, not the high season of the
occurrence of plasma irregularity, under quiet and disturbed conditions, respectively.
Observations from multi-satellites are used to study the possible factors accounting
for the occurrence of plasma irregularity. The main remarks can be summarized as
below,
1) On a quiet day, 17 August 2003, after local sunset, the plasma bubbles in 180°E
sector were detected by GRACE, ROCSAT-1 and DMSP F15 satellites. After about
100 minutes, the plasma blobs in 170°E sector were detected by ROCSAT-1 in low-
latitude region due to the westward motion of plasma irregularities.
On 18 August 2003, during the main phase of the storm, the plasma bubbles in 180°E
sector were firstly recorded, and the plasma blobs in 170°E sector were also detected
after about 100 minutes by ROCSAT-1.
2) Observations from CHAMP and GRACE indicated that EIAs were enhanced
significantly before the occurrence of plasma bubbles on the two successive days with
respect to that on other days. EIA asymmetry also displayed remarkable variations.
3) [O/N$_2$] ratio also showed the increase on 17 and 18 August 2003. The increase can
be attributed to the downward wind, generating from the enhancement of EIA
strength.
4) The remarkable enhancement of EIA strength under quiet condition can be
attributed to the enhancement post-sunset eastward electric field, due to the factors
from below, such as the gravity waves at the lower atmosphere, which need to be
further studied. In result, the enhanced EIA give rise to a downward wind in
equatorial region, which favor the initiation of R-T instability and occurrence of
plasma bubble. The downward wind also lead to the enhancement of [O/N$_2$] ratio.
The enhancement of post-sunset eastward electric field is suggested to be the most
important for the day-to-day development of plasma irregularity, which could lead to



the rapid rise of F-layer, EIA enhancement, and also the generation of vertical wind in
equatorial region.
5) Meridional wind plays an important role in the occurrence of the plasma blob in
low-latitude ionosphere. Under the effects of the meridional neutral wind, in addition
to the polarization electric field from the occurrence of plasma bubbles, the plasma
blobs occurred on two successive days.

*Acknowledgements.* The work is supported by National Natural Science Foundation
of China (41474134, 41474135, 41704161). We acknowledge UCAR, NCU, UT at
Dallas, ACE science centre, JHU for providing the satellites data and GUVI [O/N$_2$]
data, respectively.
*Data availability*. The *Dst* and $K_p$ data are downloaded from World Data Center at
Kyoto (http://swdcdb.kugi.kyoto-u.ac.jp/dstdir/index.html). The ROCSAT-1 data can
be downloaded from NCU (http://sdbweb.ss.ncu.edu.tw/ipei_download.html). The
DMSP data can be downloaded from the Center for Space Science at the University of
Texas at Dallas (http://cindispace.utdallas.edu/DMSP/dmsp_data_at_utdallas.html).
The CHAMP and GRACE data can be downloaded from CDAAC (http://cdaac-
www.cosmic.ucar.edu/cdaac/products.html). The Interplanetary Magnetic Field and
Solar wind speed are downloaded from the ACE center
(http://www.srl.caltech.edu/ACE/ASC/). The GUVI [O/N$_2$] ratio are download from
http://guvitimed.jhuapl.edu/data_on2_info. The real-time model of the ionospheric
electric field can be accessed from CIRES
(http://geomag.org/models/PPEFM/RealtimeEF.html).
*Author contribution.* WL analyzed the data and prepared the manuscript, CX
discussed the results and modified the manuscript, ZZ, SC and XY help to analyze the
data and discussed the results. All the authors read and approved the final manuscript.
*Competing interests.* The authors declare that they have no conflict of interest.

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



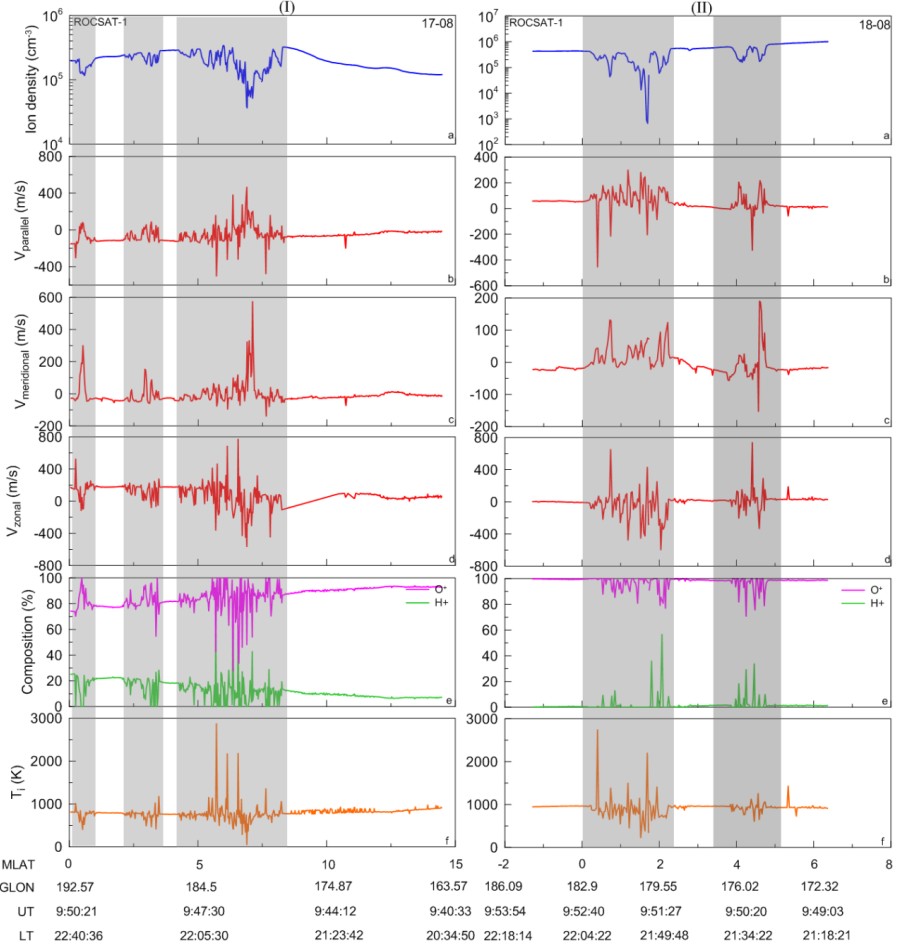

**Figure 1.** The ion density, drift, composition and temperature observed from ROCSAT-1 along the satellite trajectories in 170°E sector during 0940-0950 UT on 17 August 2003 (left panel, I) and during 0949-0952 UT on 18 August 2003 (right panel, II), respectively. $V_{parallel}$ is the field-aligned component, $V_{meridional}$ is meridional, $V_{zonal}$ is zonal.



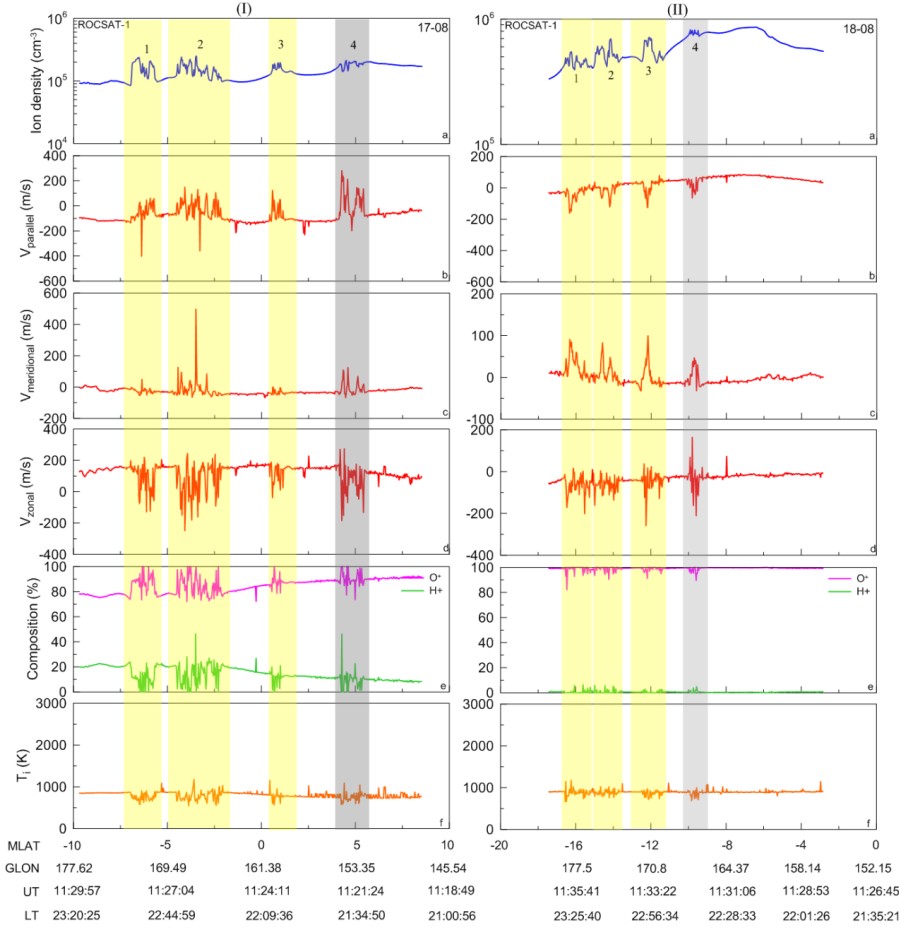

**Figure 2.** The ion density, drift, composition and temperature observed from ROCSAT-1 along the satellite trajectories in 170°E sector during 1118-1129 UT on 17 August 2003 (left panel, I) and during 1128-1135 UT on 18 August 2003 (right panel, II), respectively.

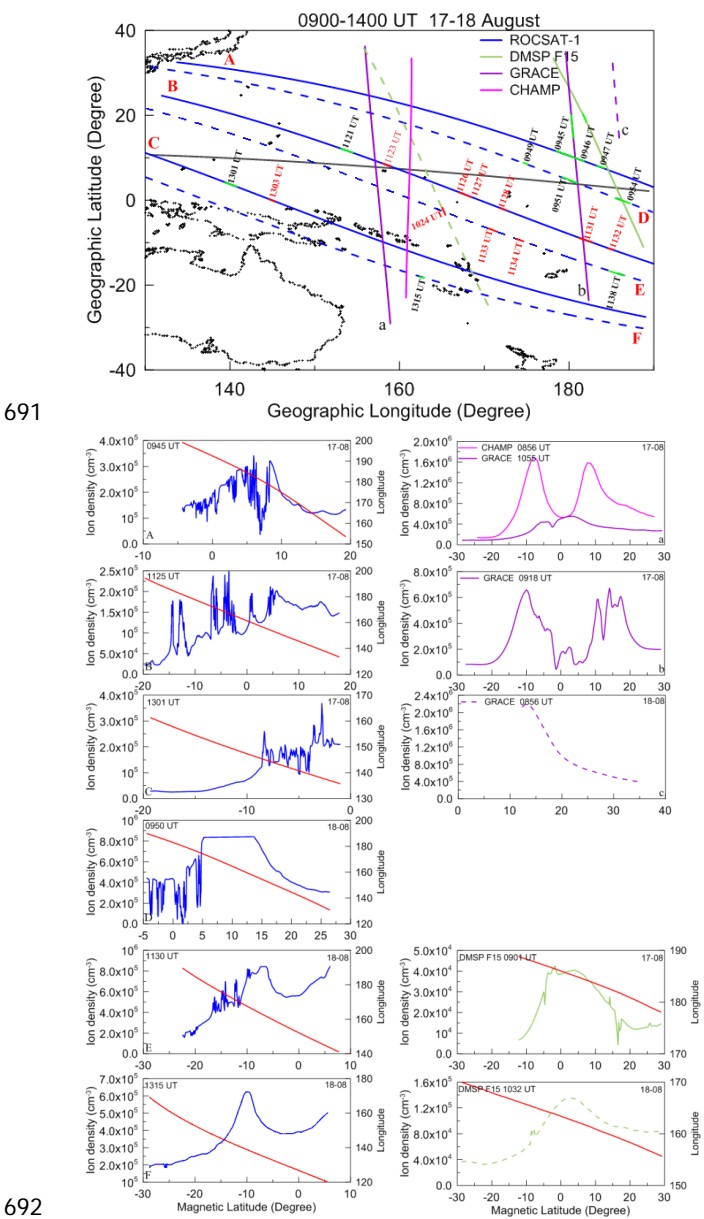

**Figure 3.** During 0900-1400 UT, on 17 and 18 August, the regional map at 130-190°E longitude sector including the trajectories of DMSP, ROCSAT-1, GRACE, CHAMP satellites (top panel), and the variations of ion density along the satellite trajectories (bottom panel). The red and green short lines represent the density enhancements and the density depletions recorded by the satellites, respectively. The solid lines represent the observations on 17 August, while the dashed lines represent the observations on 18 August.

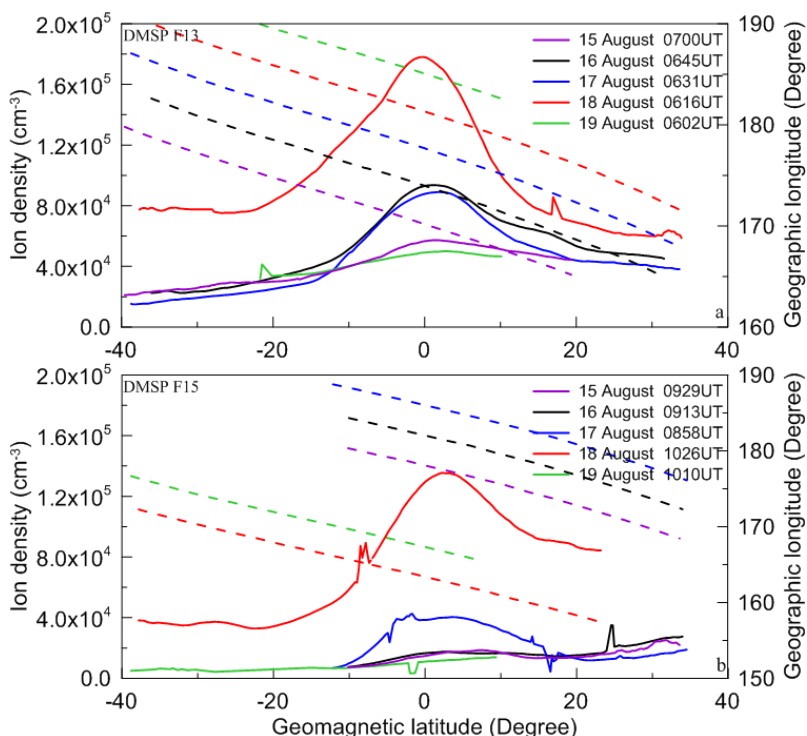


**Figure 4.** Variations of ion density and longitudinal tracks from DMSP F13 (a) and
F15 (b) along the satellite trajectories during 15-19 August 2003. The solid lines
represent the ion density, the dashed lines represent the longitudinal tracks.


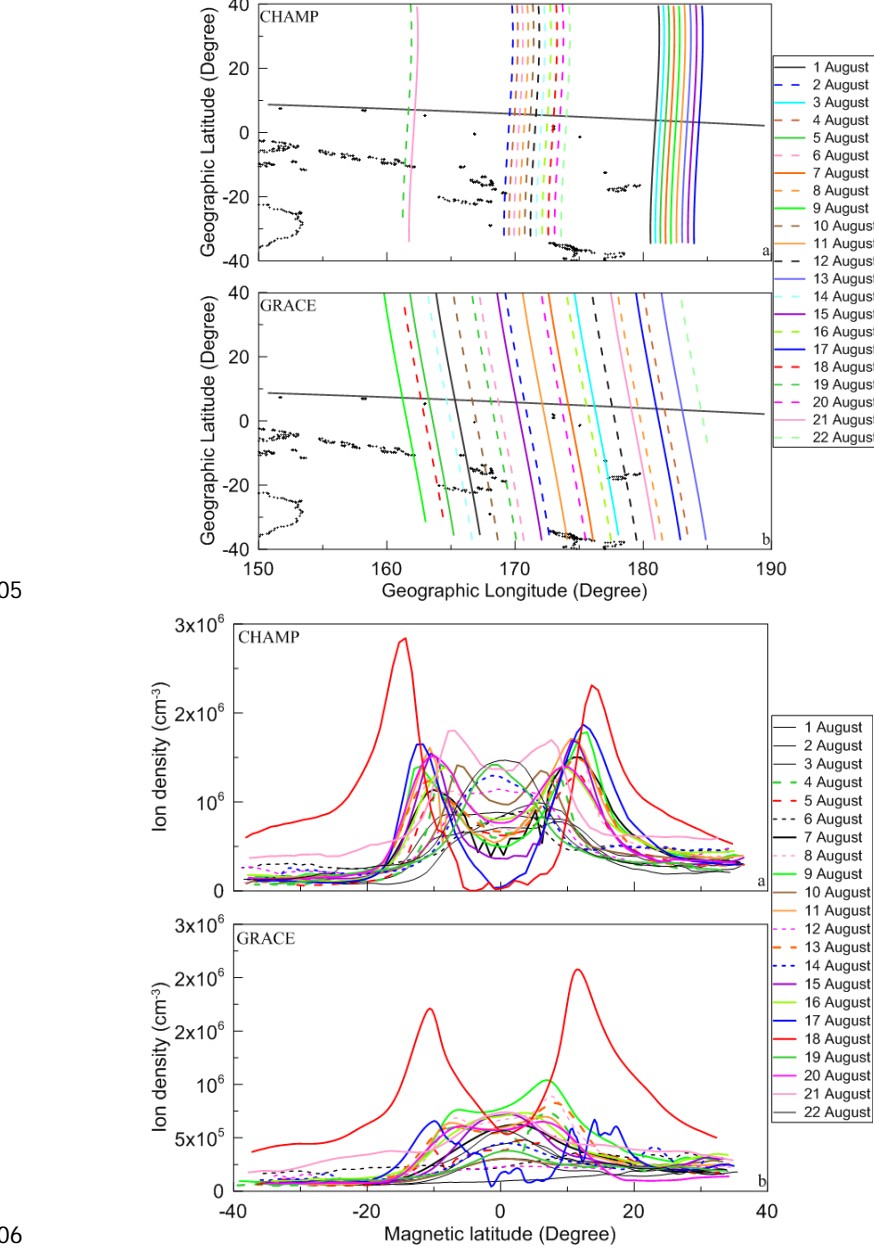



**Figure 5.** During 1-22 August 2003, in 170°E region, the tracks of CHAMP (a) and GRACE (b) satellites (top panel) and the variations of in-situ measurement of ion density (bottom panel) along the CHAMP (a) and GRACE (b) trajectories






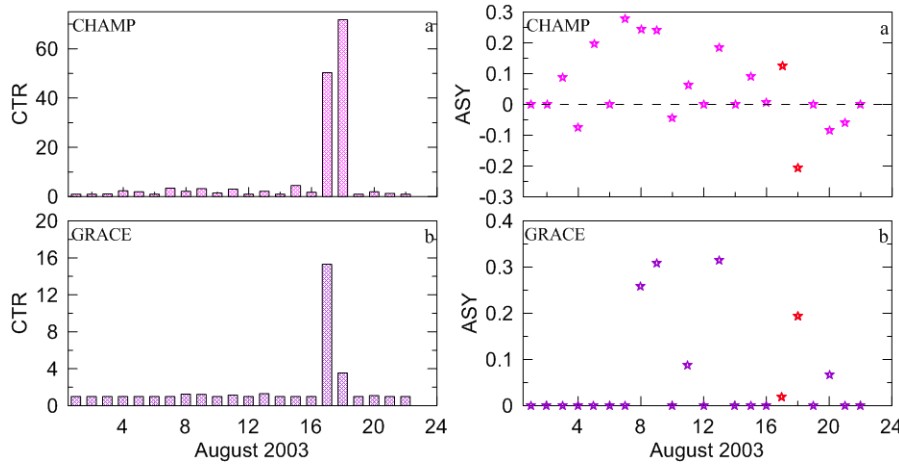

**Figure 6.** Variations of EIA strength (left panel) and asymmetry (right panel) during 1-22 August 2003 derived from CHAMP (a) and GRACE (b) observations in 170°E region



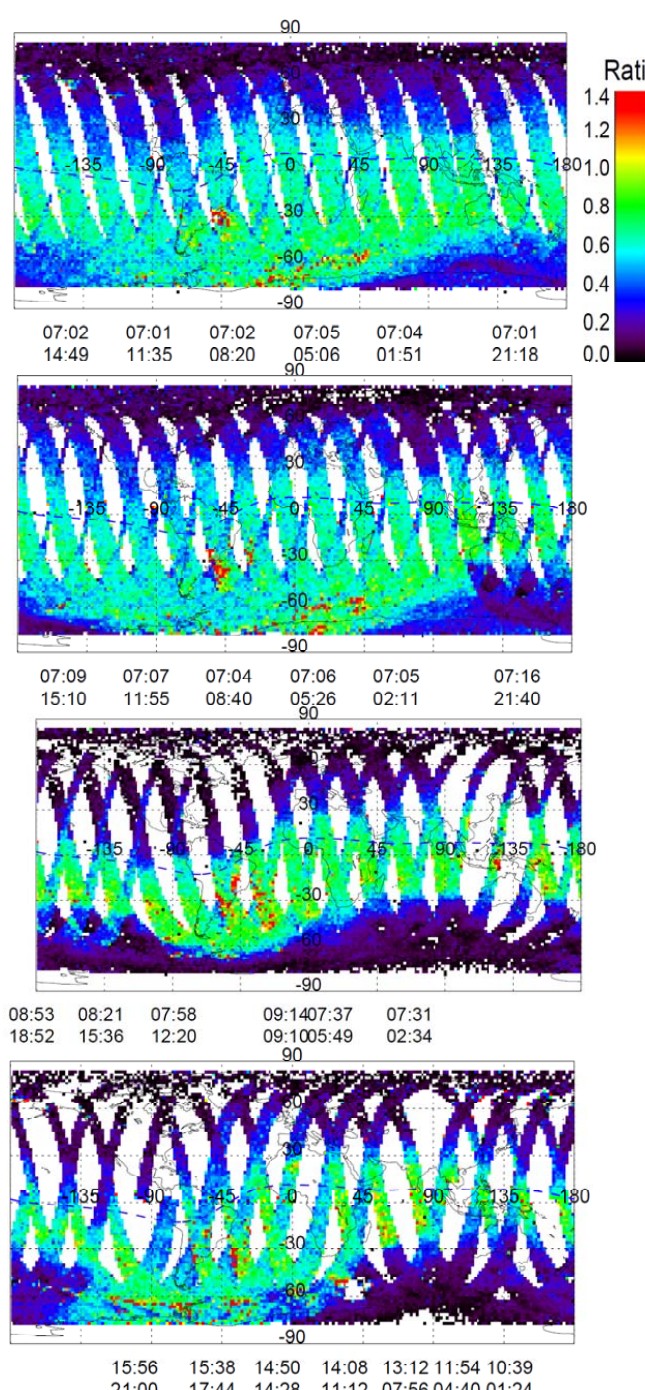

**Figure 7.** Variations of [O/N$_2$] ratio during 16-19 August 2003. From the top panel to the bottom panel, the figures represent the observations on 16 August and 19 August, respectively





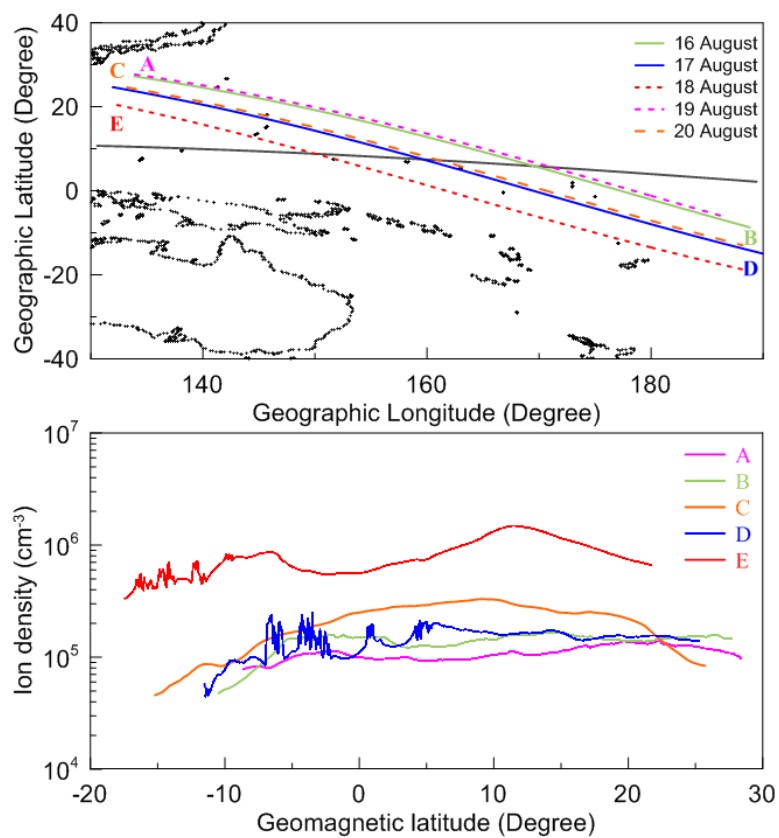

716

**Figure 8.** During 16-20 August 2003, the tracks (top panel) and density variations
(bottom panel) along the ROCSAT-1 trajectories in close tracks

719



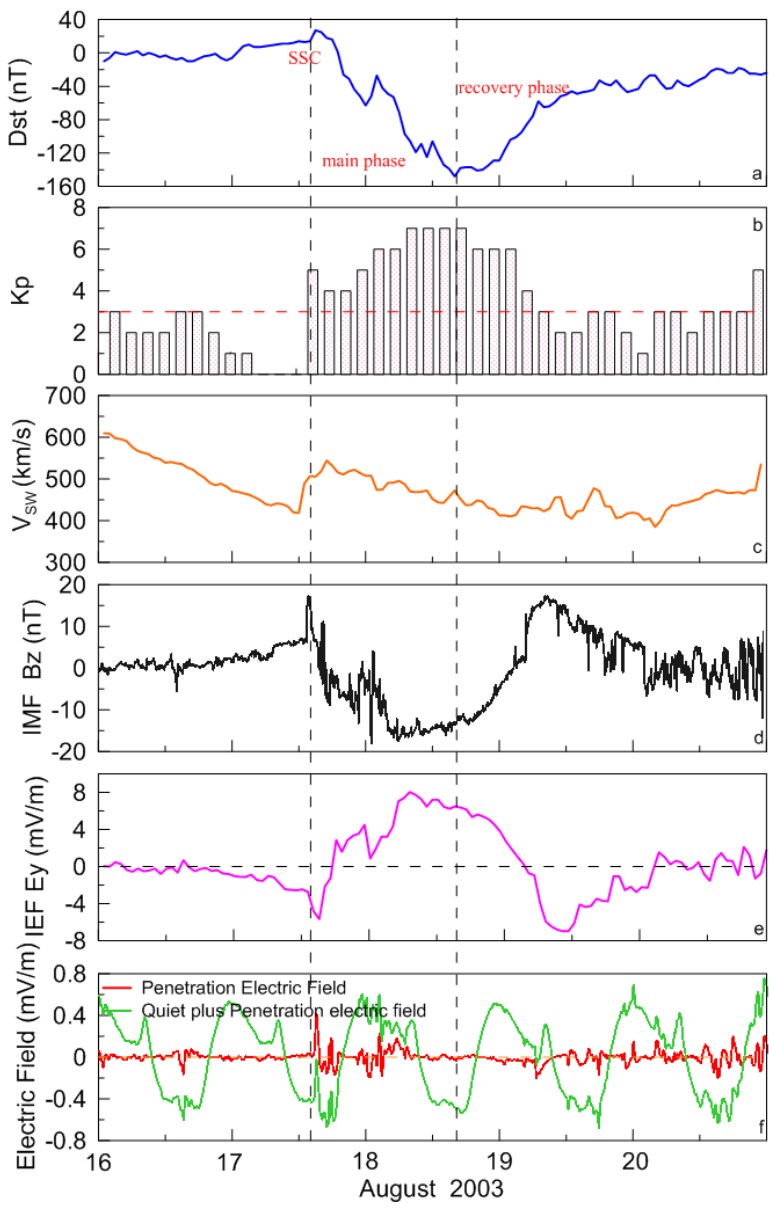

720

**Figure 9.** During 16-20 August, 2003, variations of the *Dst* index (a), *K$_p$* index (b), solar wind speed *V$_{SW}$* (c), Interplanetary Magnetic Field (IMF) *B$_z$* component (d), Interplanetary Electric Field *E$_y$* (e), and the variations of Penetration Electric Field and quiet plus penetration electric field derived from the real-time model of the ionospheric electric fields (f)

726