# Peer review of "The observations from ROCSAT-1 on 16 and 19 August 2003"

_Annales Geophysicae, 2019_

## Referee Comment (RC1) · Anonymous Referee #1 · 14 Nov 2019

Manuscript: ANGEO-2019-128 by Weihua Luo et al. This paper investigated plasma bubbles and blobs in low latitudes and the roles of electric field, neutral winds, and neutral composition in the creation of them using various satellite observations. Reading the text and following the figures required a painful effort. This paper was like a department store with various miscellaneous stuffs. Results were a simple display of various observational data, and discussion was full of speculation and repetition of known facts. I did not find any scientific value from results and discussion.

Below are Conclusions of the paper. My response to these conclusions will be good enough.

[Figure]

(1) On a quiet day, 17 August 2003, after local sunset, the plasma bubbles in 180°E sector were detected by GRACE, ROCSAT-1 and DMSP F15 satellites. After about 100 minutes, the plasma blobs in 170°E sector were detected by ROCSAT-1 in low latitude region due to the westward motion of plasma irregularities. On 18 August 2003, during the main phase of the storm, the plasma bubbles in 180°E sector were firstly recorded, and the plasma blobs in 170°E sector were also detected after about 100 minutes by ROCSAT-1. (2) Observations from CHAMP and GRACE indicated that EIAs were enhanced significantly before the occurrence of plasma bubbles on the two successive days with respect to that on other days. EIA asymmetry also displayed remarkable variations.

: These two are just the description of observational data. I do not find any scientific message from these descriptions.

(3) [O/N2] ratio also showed the increase on 17 and 18 August 2003. The increase can be attributed to the downward wind, generating from the enhancement of EIA strength.

: The O/N2 ratio provided by GUVI does not purely represent thermospheric conditions. Because the radiative recombination of oxygen ions enhances the OI 135.6 nm emission, the O/N2 ratio enhancement can be caused by the enhancement of the oxygen ion density. Without any evidence, the argument of the connection of the O/N2 ratio change to vertical winds and vertical winds to the EIA strength are not a meaningful speculation.

(4) The remarkable enhancement of EIA strength under quiet condition can be attributed to the enhancement post-sunset eastward electric field, due to the factors from below, such as the gravity waves at the lower atmosphere, which need to be further studied. In result, the enhanced EIA give rise to a downward wind in equatorial region, which favor the initiation of R-T instability and occurrence of plasma bubble. The downward wind also lead to the enhancement of [O/N2] ratio. The enhancement of post-sunset eastward electric field is suggested to be the most important for the dayto-day development of plasma irregularity, which could lead to the rapid rise of F-layer, EIA enhancement, and also the generation of vertical wind in equatorial region.

: I do not know how gravity waves can produce such a strong eastward electric fields. All the descriptions here are based on speculation which has no scientific value.

(5) Meridional wind plays an important role in the occurrence of the plasma blob in low-latitude ionosphere. Under the effects of the meridional neutral wind, in addition to the polarization electric field from the occurrence of plasma bubbles, the plasma blobs occurred on two successive days.

: The authors do not have the measurements of meridional winds. Blobs can be produced by the mechanism that the authors mentioned here, but this statement does not provide any constructive answer to the question of the origin of blobs.

---

## Referee Comment (RC2) · Anonymous Referee #2 · 18 Nov 2019

The paper studied the occurrence of ionospheric plasma irregularities (bubbles and blobs) on days 17 and 18 of August 2003 over the longitude of 170o using data of the ROCSAT-1, DMSP, Grace and CHAMP satellites. They successfully detected these irregularities as signatures on the ion density measured by these satellites and they also used the ratio [O/N2] from GUVI to support the presence of vertical neutral wind. Even though the authors presented many discussions about the physical mechanisms responsible for the irregularities they need to consider the following points/suggestions: - the authors presented many well-known (for decades) physical mechanisms and conclusions; - the paper doesn't present any new insight and data do not support the suggestions for these mechanisms; - the authors observed (line 157) westward bubble

movements for the quiet (17 th) and for the disturbed day (18 th). Normally the bubbles drift to east during magnetically quiet days and this movements reverts to west during disturbed days. How the authors explain this? - the authors observed plasma irregularities on both quiet (17 th) and disturbed day (18 th). For the quiet day they suggest that waves (LSWS, PW, GW) propagating upward (from down) contributed to trigger the irregularities while during the disturbed day they attributed the PPEF as the main cause for the irregularities generation. They pointed out that there were no irregularities on the other days of August 2003 (non-irregularity season at that longitudinal sector). Why these upward waves were so active on day 17 and not at the other days of August? No support for these upward waves is presented. - at Figure 4, using DMSP data at 840 km large ion density is observed on day 18 at geomagnetic equator while large ion densities are observed at the EIA crests (see Figure 5 , CHAMP and Grace data) with no ionization at magnetic equator. Even though the altitudes of DMSP and CHAMP/GRACE satellites are different it is expected that high density peaks at the EIA crests comes from magnetic equator. How authors explain this? Here also another comment: there are differences between geomagnetic equator (Figure 4) and magnetic equator (Figure 5). The authors should standardize this or was it a typo error? -at Figure 6 why CTR is higher at day 18 compared to day 17 in the CHAMP data while CTR is higher at day 17 compared to day 18 in GRACE data if the satellite had just 90 km difference in altitude? - the authors pointed out that "the plasma bubble is preferable to occur when the EIA is symmetric" (and here some papers are referred) however on day 17 and especially on day 18 larger asymmetries (lines 230 to 232) were observed (see Figure 6) on these EIA crests when compared to days 16 and 19. As large occurrence of bubbles was observed on days 17 and 18 how the authors explain this fact? - at Figure 7 please clarify the white background. Is this lack of data coverage? - at Figure 9 is advisable to use SYM H (if available) to have more time resolution what can help to associate it's variation with the other parameters. - English should be improved at some papers (below some parts are pointed out).

Minor but not less important suggestions/comments/questions are:

Line Suggestions/comments/questions 25-27 Well known for many decades (as pointed out above) 55 The word enigma is too strong since many works were already done on this subject 71-72 Clarify 78-79 Discuss here the effect of the magnetic declination 86-88 Confused. Rewrite 112-113 Check if it is 0600 LT and 0900 LT since there are no irregularities during daytime (normally) 220 It is . . . 230 71 instead of 710 236 Instead of around please specify the latitudinal range 241 Specify here that this irregularity preferable season is for the 170o 249 nq should be lower case 252 Explain $\beta$ 276 were observed .. 278 What is the mean for "Supplement"? 284 Check signal for this equation 301 Explain from upper 318 Improve: driving the crests move to the equator 321-325 Improve 330 Same of line 301 354 while the .. 370 result that should . . . 378 EIA is enhanced 382 EIAS during days 17 and 18 were . . . 391 upper, and the . . .

---

## Author Comment (AC1) · 24 Dec 2019

Anonymous Referee #1: Interactive comment on "A case study of the day-to-day occurrence of plasma irregularities in low-latitude ionosphere from multi-satellite observations" by Weihua Luo et al. Manuscript: ANGEO-2019-128 by Weihua Luo et al. This paper investigated plasma bubbles and blobs in low latitudes and the roles of electric field, neutral winds, and neutral composition in the creation of them using various satellite observations. Reading the text and following the figures required a painful effort. This paper was like a department store with various miscellaneous stuffs. Results were a simple display of various observa-

tional data, and discussion was full of speculation and repetition of known facts. I did not find any scientific value from results and discussion. Below are Conclusions of the paper. My response to these conclusions will be good enough.

Response: We would like to thank the reviewer for her/his constructive comments and suggestions on our study. In this manuscript, we report the occurrence of the plasma bubbles and blobs in two successive days under quiet and disturbed conditions, respectively. The variations of Equatorial Ionization Anomaly (EIA) during the two days were similar. The interesting results can be summarized as, 1) The EIAs were strengthened during the quiet day (17 August) and the disturbed day (18 August), which means the zonal electric fields were enhanced. On the disturbed day, the enhancement of electric field can be attributed to the Prompt Penetration Electric Field (PPEF). On the quiet day, we prove that the enhancement was not associated with the PPEF, which may be related with the lower atmosphere, such as Large-Scale Wave Structure (LSWS). 2) The EIAs became asymmetric in these two days, which may be related with the meridional wind. Generally, the plasma bubble is preferable to occur when the EIA is symmetric because of the absent of the meridional wind. When the co-existence of enhanced eastward electric field and meridional wind, the competition of the two factors should be considered. On 17 and 18 August, the eastward electric fields were enhanced and lead to the occurrence of irregularities, though the EIA became asymmetric due to the existence of meridional wind. Furthermore, the enhanced eastward electric field will strengthen the EIA, and the occurrence of vertical wind in equatorial region, which are preferable to the generation of plasma bubbles. From the results, we can speculate that the effects of enhancement of eastward electric field play the first important role in the occurrence of irregularities, while meridional wind may play the secondary important role in the occurrence of irregularities. The factors leading to the enhancement of eastward electric field under quiet condition need to be further investigated, which may be the key factor for the day-to-day occurrence of plasma irregularities in equatorial and low-latitude ionosphere.

(1) On a quiet day, 17 August 2003, after local sunset, the plasma bubbles in 180_E sector were detected by GRACE, ROCSAT-1 and DMSP F15 satellites. After about 100 minutes, the plasma blobs in 170_E sector were detected by ROCSAT-1 in low latitude region due to the westward motion of plasma irregularities. On 18 August 2003, during the main phase of the storm, the plasma bubbles in 180_E sector were firstly recorded, and the plasma blobs in 170_E sector were also detected after about 100 minutes by ROCSAT-1. (2) Observations from CHAMP and GRACE indicated that EIAs were enhanced significantly before the occurrence of plasma bubbles on the two successive days with respect to that on other days. EIA asymmetry also displayed remarkable variations. : These two are just the description of observational data. I do not find any scientific message from these descriptions.

Response: We summarize the observational results here. The report describes that the plasma bubbles and blobs were detected in the same region on two successive days, and EIAs were enhanced significantly and became asymmetric before the occurrence of plasma bubbles. These descriptions provide the foundation to discuss the factors leading to the occurrence of irregularities.

(3) [O/N2] ratio also showed the increase on 17 and 18 August 2003. The increase can be attributed to the downward wind, generating from the enhancement of EIA strength. : The O/N2 ratio provided by GUVI does not purely represent thermospheric conditions. Because the radiative recombination of oxygen ions enhances the OI 135.6 nm emission, the O/N2 ratio enhancement can be caused by the enhancement of the oxygen ion density. Without any evidence, the argument of the connection of the O/N2 ratio change to vertical winds and vertical winds to the EIA strength are not a meaningful speculation.

Response: As the reviewer pointed out, the [O/N2] ratio enhancement can also be caused by the enhancement of the oxygen ion density. On 18 August, the O+ inside the plasma bubbles recorded by ROCSAT-1 showed the decrease. It is possible to say that the increase of [O/N2] ratio was from the downward wind, generating from the
enhancement of EIA. (4) The remarkable enhancement of EIA strength under quiet condition can be attributed to the enhancement post-sunset eastward electric field, due to the factors from below, such as the gravity waves at the lower atmosphere, which need to be further studied. In result, the enhanced EIA give rise to a downward wind in equatorial region, which favor the initiation of R-T instability and occurrence of plasma bubble. The downward wind also lead to the enhancement of [O/N2] ratio. The enhancement of post-sunset eastward electric field is suggested to be the most important for the day-to-day development of plasma irregularity, which could lead to the rapid rise of F-layer, EIA enhancement, and also the generation of vertical wind in equatorial region. : I do not know how gravity waves can produce such a strong eastward electric fields. All the descriptions here are based on speculation which has no scientific value.

Response: On 17 August, a quiet day, the EIA was strengthened and became asymmetric, which means the existence of enhanced eastward electric field and meridional wind. The results from the calculation of Interplanetary Electric Field (IEF) and Prompt Penetration Electric Field (PPEF) model showed that the background zonal electric field was not affected by the factors from the upper, e.g. PPEF. The factors from the lower atmosphere may also enhance the F-region electric field. Thus, we speculate that the background zonal electric field was strengthened due to the Large-Scale Wave Structure (LSWS), which may be a key factor to the day-to-day occurrence of irregularities in equatorial and low-latitude ionosphere. In fact, we also do not know how the waves can strongly affect the electric field (upward plasma drift) on 17 August, and we want to study the effects of the LSWS on the zonal electric field in F-region. Unfortunately, we cannot access any observational data or model simulation to prove this speculation. The effects of LSWS on the F-region plasma drift or zonal electric field need to be further studied.

(5) Meridional wind plays an important role in the occurrence of the plasma blob in low latitude ionosphere. Under the effects of the meridional neutral wind, in addition

to the polarization electric field from the occurrence of plasma bubbles, the plasma blobs occurred on two successive days. : The authors do not have the measurements of meridional winds. Blobs can be produced by the mechanism that the authors mentioned here, but this statement does not provide any constructive answer to the question of the origin of blobs.

Response: In the manuscript, we do not have the measurements of meridional winds. It is difficult to measure the meridional winds at some longitudinal regions, such as the 170°E in this study. Thus, we attempt to analyze physically the possible factors leading to the occurrence of plasma bubbles and blobs, from the variations of Equatorial Ionization Anomaly (EIA). It is well known that the major factors affecting the development, strength and asymmetry of EIA are zonal electric field and meridional wind (e.g. Balan et al., 2018; Khadka et al., 2018). Based on the observations from CHAMP and GRACE, which can be well used to study the latitudinal variations of electron density, the results showed that the EIAs on the two successive days (17 and 18 August) were strengthened and became asymmetric, which means that the co-existence of enhanced eastward electric field and meridional wind in physics. Under the presence of meridional wind, the plasma blob may generate after the occurrence of plasma bubbles.

References Balan, N., L. Liu, and H. Le (2018), A brief review of equatorial ionization anomaly and ionospheric irregularities, Earth and Planetary Physics, 2: 2570275, doi:10.26464/epp2018025 Khadka, S.M., C.E. Valladares, R. Sheehan, A.J. Gerrard (2018), Effects of electric field and neutral wind on the asymmetry of equatorial ionization anomaly, Radio Science, 53,683-697

Please also note the supplement to this comment:
https://www.ann-geophys-discuss.net/angeo-2019-128/angeo-2019-128-AC1-supplement.pdf

---

## Author Comment (AC2) · 24 Dec 2019

Anonymous Referee #2: Interactive comment on "A case study of the day-to-day occurrence of plasma irregularities in low-latitude ionosphere from multi-satellite observations" by Weihua Luo et al.

The paper studied the occurrence of ionospheric plasma irregularities (bubbles and blobs) on days 17 and 18 of August 2003 over the longitude of 170o using data of the ROCSAT-1, DMSP, Grace and CHAMP satellites. They successfully detected these irregularities as signatures on the ion density measured by these satellites and they also used the ratio [O/N2] from GUVI to support the presence of vertical neutral wind.

Even though the authors presented many discussions about the physical mechanisms responsible for the irregularities they need to consider the following points/suggestions:

Response: We would like to thank the reviewer for her/his invaluable comments and suggestions on our study.

- the authors presented many well-known (for decades) physical mechanisms and conclusions;- the paper doesn't present any new insight and data do not support the suggestions for these mechanisms;

Response: In this manuscript, we report the occurrence of the plasma bubbles and blobs in two successive days under quiet and disturbed conditions, respectively. The variations of Equatorial Ionization Anomaly (EIA) during the two days were similar. The interesting results can be summarized as, 1) The EIAs were strengthened during the quiet day (17 August) and the disturbed day (18 August), which means the zonal electric fields were enhanced. On the disturbed day, the enhancement of electric field can be attributed to the Prompt Penetration Electric Field (PPEF). On the quiet day, we prove that the enhancement was not associated with the PPEF, which may be related with the lower atmosphere, such as Large-Scale Wave Structure (LSWS). 2) The EIAs became asymmetric in these two days, which may be associated with the meridional wind. Generally, the plasma bubble is preferable to occur when the EIA is symmetric because of the absent of the meridional wind. When the co-existence of enhanced eastward electric field and meridional wind, the competition of the two factors should be considered. On 17 and 18 August, the eastward electric fields were enhanced and lead to the occurrence of irregularities, though the EIA became asymmetric due to the existence of meridional wind. Furthermore, the enhanced eastward electric field will strengthen the EIA, and the occurrence of vertical wind in equatorial region, which are preferable to the generation of plasma bubbles. From the results, we can speculate that the effects of enhancement of eastward electric field play the first important role in the occurrence of irregularities, while meridional wind may play the secondary important role in the occurrence of irregularities. The factors leading to the enhancement of

eastward electric field under quiet condition need to be further investigated, which may be the key factor for the day-to-day occurrence of plasma irregularities in equatorial and low-latitude ionosphere.

- the authors observed (line 157) westward bubble movements for the quiet (17th) and for the disturbed day (18th). Normally the bubbles drift to east during magnetically quiet days and this movements reverts to west during disturbed days. How the authors explain this?

Response: As the reviewer pointed out, the bubbles usually move to east during magnetically quiet days and reverts to west during disturbed days. It's our carelessness that the movements of the bubbles were not discussed in the manuscript. In Figure 1 and Figure 2, the plasma inside the irregularities moved westward and the ambient plasma moved eastward. The drifts satellite detected were the plasma drifts, not the bubble drifts. As Huang et al. (2010) reported, the zonal drift velocity of the plasma particles inside plasma bubbles is significantly different from the ambient plasma drift. The relative zonal velocity of the ions inside the depletion region with respect to the ambient plasma is generally westward. The drifts shown in Figure 1 and Figure 2 are normal and reasonable.

Reference

Huang, C.S., O. de La Beaujardiere, R.F. Pfaff, J.M. Retterer, P.A. Roddy, D.E. Hunton, Y.J. Su, S.Y. Su, F.J. Rich (2010), Zonal drift of plasma particles inside equatorial plasma bubbles and its relation to the zonal drift of the bubble structure, Journal of Geophysical Research: Space Physics, 115(A7), A07316

- the authors observed plasma irregularities on both quiet (17 th) and disturbed day (18 th). For the quiet day they suggest that waves (LSWS, PW, GW) propagating upward (from down) contributed to trigger the irregularities while during the disturbed day they attributed the PPEF as the main cause for the irregularities generation. They pointed out that there were no irregularities on the other days of August 2003 (non-irregularity

season at that longitudinal sector). Why these upward waves were so active on day 17 and not at the other days of August? No support for these upward waves is presented.

Response: On 17 August, a quiet day, the EIA was strengthened and became asymmetric, which means the existence of enhanced eastward electric field and meridional wind. The background zonal electric field was not affected by the factors from the upper, e.g. penetration electric field. The factors from the lower atmosphere may also enhance the F-region electric field. Thus, we speculate that the background zonal electric field was strengthened due to the Large-Scale Wave Structure (LSWS), which may be a key factor to the day-to-day occurrence of irregularities in equatorial and low-latitude ionosphere. In fact, we also do not know why the waves were so active on 17 August. We want to study the effects of the LSWS on the zonal electric field in F-region. Unfortunately, we cannot access any observational data or model simulation to prove this speculation.

- at Figure 4, using DMSP data at 840 km large ion density is observed on day 18 at geomagnetic equator while large ion densities are observed at the EIA crests (see Figure 5 , CHAMP and Grace data) with no ionization at magnetic equator. Even though the altitudes of DMSP and CHAMP/GRACE satellites are different it is expected that high density peaks at the EIA crests comes from magnetic equator. How authors explain this? Here also another comment: there are differences between geomagnetic equator (Figure 4) and magnetic equator (Figure 5). The authors should standardize this or was it a typo error?

Response: On 18 August, the ion densities at the equator and EIA crests were large at 400 km and 840 km. Due to the "fountain effect", the plasma at the higher altitude and lower latitude (equatorial region) would diffuse along the field line to the lower altitude and higher latitude. At DMSP altitude (840 km), if the EIA is well developed (high density peaks at the EIA crests), the plasma should be transported to very high height at the equatorial region, which is difficult. For instance, if the EIA crests locate at about $\pm10°$N, the apex of the field line is about 1064 km. Though the ion density

in equatorial region at DMSP altitude was large, lower density at the EIA crests was normal and reasonable. It is a typo. The coordinate systems in Figure 4 and Figure 5 are geomagnetic.

-at Figure 6 why CTR is higher at day 18 compared to day 17 in the CHAMP data while CTR is higher at day 17 compared to day 18 in GRACE data if the satellite had just 90 km difference in altitude? - the authors pointed out that "the plasma bubble is preferable to occur when the EIA is symmetric" (and here some papers are referred) however on day 17 and especially on day 18 larger asymmetries (lines 230 to 232) were observed (see Figure 6) on these EIA crests when compared to days 16 and 19. As large occurrence of bubbles was observed on days 17 and 18 how the authors explain this fact?

Response: In Figure 6, the variations of EIA strength (CTR) and asymmetry (ASY) were different at CHAMP and GRACE altitude. From the variations of the EIA, we can speculate the existence of meridional wind, which may also affect the strength of EIA. The meridional wind may have the altitudinal and latitudinal gradient (Huba and Krall, 2013; Meriwether et al., 2008) , thus the EIA strength and asymmetry were different at different altitudes. Generally, the plasma bubble is preferable to occur when the EIA is symmetric due to the effect of meridional wind. When the co-existence of enhanced eastward electric field and meridional wind, the competition of the two factors would be considered. On 17 and 18 August, the eastward electric fields were enhanced and lead to the occurrence of irregularities, though the EIA became asymmetric due to the meridional wind. It also means that the effects of enhancement of eastward electric field play an important role in the occurrence of irregularities, while meridional wind may not play the first important role in the occurrence of irregularities.

Reference

Huba, J.D., and J. Krall (2013), Impact of meridional winds on equatorial spread F: Revisited, Geophysical Research Letters, 40, 1268-1272

[Figure]

Meriwether, J., M. Faivre, C. Fesen, P. Sherwood, and O. Veliz (2008), New results on equatorial thermospheric winds and the midnight temperature maximum, Ann. Geophysicae, 26, 447

- at Figure 7 please clarify the white background. Is this lack of data coverage?

Response: Figure 7 is downloaded from the website (http://guvitimed.jhuapl.edu/data_products) , the white background is the lack of data coverage.

- at Figure 9 is advisable to use SYM H (if available) to have more time resolution what can help to associate it0s variation with the other parameters. – English should be improved at some papers (below some parts are pointed out).

Response: Thanks very much for the useful suggestions. We would try to study the variations of SYM-H index and improve our writing in the text.

Minor but not less important suggestions/comments/questions are:

Line Suggestions/comments/questions 25-27 Well known for many decades (as pointed out above) 55 The word enigma is too strong since many works were already done on this subject 71-72 Clarify 78-79 Discuss here the effect of the magnetic declination 86-88 Confused. Rewrite 112-113 Check if it is 0600 LT and 0900 LT since there are no irregularities during daytime (normally) 220 It is : : : 230 71 instead of 710 236 Instead of around please specify the latitudinal range 241 Specify here that this irregularity preferable season is for the 170o 249 nq should be lower case 252 Explain _ 276 were observed ..

Response: Thanks very much for useful suggestions.

278 What is the mean for "Supplement"?

Response: We plot the observations on 16 and 19 August 2003 from ROCSAT-1 as the "Supplement", to demonstrate there were no irregularities on other days.
284 Check signal for this equation 301 Explain from upper 318 Improve: driving the crests move to the equator 321-325 Improve 330 Same of line 301 354 while the .. 370 result that should: : : 378 EIA is enhanced 382 EIAS during days 17 and 18 were : : : 391 upper, and the : : :

Response: Thanks very much for useful suggestions.

Please also note the supplement to this comment:
https://www.ann-geophys-discuss.net/angeo-2019-128/angeo-2019-128-AC2-supplement.pdf